# Systemic *Candida* Infection and Pulmonary Aspergillosis in an Alpaca (*Vicugna pacos*): A Case Report

**DOI:** 10.3390/jof10030227

**Published:** 2024-03-20

**Authors:** Andrea Grassi, Claudia Cafarchia, Nicola Decaro, Wafa Rhimi, Vittoriana De Laurentiis, Giulia D’Annunzio, Andrea Luppi, Paola Prati

**Affiliations:** 1Istituto Zooprofilattico Sperimentale della Lombardia e dell’Emilia-Romagna, 27100 Pavia, Italy; andrea.grassi@izsler.it (A.G.); paola.prati@izsler.it (P.P.); 2Dipartimento di Medicina Veterinaria, Università degli Studi di Bari, 70100 Bari, Italy; nicola.decaro@uniba.it (N.D.); wafa.rhimi@uniba.it (W.R.); 3UOC Microbiology and Virology, Azienda Ospedaliera-Universitaria, Policlinico di Bari, 70100 Bari, Italy; vittoriana.delaurentiis@gmail.com; 4Istituto Zooprofilattico Sperimentale della Lombardia e dell’Emilia-Romagna, 41100 Modena, Italy; giulia.dannunzio@izsler.it; 5Istituto Zooprofilattico Sperimentale della Lombardia e dell’Emilia-Romagna, 43100 Parma, Italy; andrea.luppi@izsler.it

**Keywords:** *Candida albicans*, *Aspergillus fumigatus*, *Vicugna pacos*, *Candidemia*, *Aspergillosis*

## Abstract

This study reports a peculiar case of systemic candidiasis infection associated with pulmonary aspergillosis in an apparently immunocompetent alpaca. A captive 7-year-old female alpaca exhibited respiratory symptoms, underwent treatment with benzylpenicillin and dexamethasone, and succumbed to the infection 40 days later. During the post-mortem examination, subcutaneous emphysema, widespread pneumonia with multiple suppurative foci, scattered necro-suppurative lesions throughout the renal and hepatic parenchyma were evident. Histopathological analysis of the collected tissues revealed multifocal mild lymphoplasmacytic chronic interstitial nephritis, necro-suppurative pneumonia with the presence of fungal hyphae, multifocal foci of mineralization, and fibrosis in the liver. Fungal cultures confirmed the growth of *Aspergillus fumigatus* from the lungs, and *Candida albicans* from the liver, kidney, and heart. The only recognizable risk factor for candidiasis and pulmonary aspergillosis in this case was prior corticosteroid and antibiotic therapy. Nevertheless, it is crucial to consider systemic candidosis and pulmonary aspergillosis as potential differential diagnoses in respiratory infections among camelids. Prolonged treatment with glucocorticoids and antibiotics should be avoided as it could represent a risk factor for the onset of pathologies caused by opportunistic fungi such as *Candida* spp. and *Aspergillus* spp.

## 1. Introduction

Fungal infections pose a threat to animal health due to their high mortality and morbidity rates [1]. They can be classified as superficial, subcutaneous, and systemic infections. Systemic mycosis are rarely reported in animals, but when they occur, they are most commonly caused by *Candida* spp. and *Aspergillus* spp. *Candida* species are widely distributed yeasts and well recognized as commensal organisms of human/animal skin and/or gastrointestinal and urogenital tracts. The genus comprises more than 200 species, but only a few species have been identified as pathogens [2,3,4,5]. In particular, *Candida albicans* is commonly identified as the causative agent of candidiasis, a condition observed mainly in human or animals with compromised immune system or underlying disorders [3,6]. Localized candidiasis is common in captive wildlife, notably in avian species and mammals [7,8], whereas disseminated forms of the illness are less commonly described. The dissemination of *Candida* spp. varies depending on the host’s immune status; the yeasts can remain localized to the skin and mucous membranes, causing superficial mycoses in immunocompetent host or can spread through the bloodstream, leading to systemic mycosis in severe immunocompromised host [2,9]. Systemic candidiasis is rarely reported in animals, and only few cases were described in dogs [10,11,12,13], calves [14], sheep [15,16], cat [17], horses [18], guanaco [19], llama [20], and alpaca [21]. On the other hand, *Aspergillus* spp. are opportunistic environmental filamentous fungi, capable of causing localized respiratory issues or severe, life-threatening invasive infections [22]. They primarily infect anatomical cavities such as ear, nose, and paranasal sinuses [23]. Although *Aspergillus* genus comprises several hundred species, only a limited number of them significantly impact human and animal health. Infections commonly arise from species belonging to *A. fumigatus* complex, *A. flavus* complex, and *A. terreus* complex [24,25,26,27]. *A. fumigatus* is responsible for over 90% of infections, followed by *A. flavus* and *A. niger* in terms of frequency [28,29]. Pulmonary aspergillosis in New World camelid has been reported only in one llama [30] and in one alpaca and was caused by *A. niger* [23].

This study reports a peculiar case of systemic *Candida albicans* infection associated with pulmonary aspergillosis in an apparently immunocompetent alpaca. To the best of the authors’ knowledge, this is the first reported case of candidiasis associated with pulmonary aspergillosis in a camelid.

## 2. Case Presentation

A 7-year-old female Alpaca (*Vicugna pacos*), residing in Northern Italy in a pet therapy center alongside eleven other conspecifics, exhibited respiratory problems characterized by a distinct rattling sound resembling a crackle. During veterinary visits, the animal appeared lean, with normal body temperature, and its medical history did not reveal any health problems, pathologies, or prolonged antibiotic or corticosteroid treatments.

An empirical therapy consisting of benzylpenicillin and dexamethasone was administered for two weeks, resulting in an improvement in symptoms and a reduction in respiratory sounds.

Approximately one month after the beginning of the therapy, the condition worsened. The previously established therapy was resumed, but after one week, the animal ceased to eat and, after five days, died spontaneously.

### 2.1. Necropsy

Complete necropsy was conducted on the carcass, which unveiled the presence of subcutaneous emphysema primarily localized to the trunk and thighs, diffuse suppurative pneumonia (Figure 1), and scattered, multifocal, necrotic-purulent lesions of varying sizes and different stages of evolution throughout the renal (Figure 2) and hepatic parenchyma.

During the necropsy, samples of lung, liver, heart, and kidney were collected for microbiological and histological analysis.

### 2.2. Microbiological Investigations

Tissue samples were cultured on Sabouraud’s dextrose agar (SDA), Trypticase Soy Agar with 5% Sheep Blood and MacConkey (IZSLER) and incubated at 37 °C for 48 h.

No bacterial growth was observed while the fungal culture was positive. After 48 h of incubation, growth of cream-colored pasty colonies with a distinctive yeast smell was observed from the samples of kidney, liver, and heart, while the lung samples showed the growth of filamentous green colonies.

Gram staining was performed on the yeast colonies, which revealed the presence of Gram-positive narrow-neck budding yeast cells.

A wet mount preparation with lactophenol cotton blue was performed for the filamentous fungal colonies, revealing the presence of septate hyphae, rough-walled stipes, mature vesicles bearing phialides across the entire surface, and conspicuously echinulate conidia.

The yeast and filamentous colonies were identified with high confidence (range > 2) as *Candida albicans* and *Aspergillus fumigatus*, respectively, using a mass spectrometer MALDI-TOF Biotyper^®^ (Bruker Daltonics GmbH & Co.KG., Bremen, Germany), following the manufacturer’s instructions for fungi.

### 2.3. Molecular Identification

The identification of isolates was molecularly confirmed by the amplification and sequencing of the nuclear ribosomal internal transcribed spacer (ITS) region. Genomic DNA was isolated from each sample using the DNeasy Blood and Tissue Kit (QIAGEN, Hilden, Germany), following manufacturer’s instructions. The nuclear ribosomal ITS region was amplified using ITS1 (5’-TCCGTAGGTGAACCTGCGG-3’) and ITS4 (5’-TCCTCCGCTTATTGATATGC-3’) primers [31]. Another PCR was performed to identify *A. fumigatus* based on beta-tubulin (βtub) partial gene sequences. The primer sets βtub3 (5′-TTCACCTTCAGACCGGT-3′) and βtub2 (5′-AGTTGTCGGGACGGAATAG-3′) were used to amplify DNA fragment of the gene [32]. The PCR products were purified and sequenced in both directions using the same primers, employing the Big Dye Terminator v.3.1 chemistry in a 3130 Genetic analyzer (Applied Biosystems, Foster, CA, USA) in an automated sequencer (ABI Prism^®^ 377 DNA Sequencers, Applied Biosystems, MA, USA) Nucleotide sequences were edited, aligned, and analyzed using Bioedit sequence Alignment Editor 7.0.5.3 [33], and compared with available sequences in the GenBank using Basic Local Alignment Search Tool (BLAST; accessed on day 20 September 2023; http://blast.ncbi.nlm.nih.gov/Blast.cgi).

The BLASTn search of the Genbank database using ITS sequences as query showed 100% nucleotide identity with *C. albicans* (KY101878) and *A. fumigatus* (MH865792). Moreover, BLASTn analysis of the βtub sequence revealed the nucleotide identity of 99.8% with reference strains *A. fumigatus* from GenBank (KU897017). All sequences were deposited in the NCBI Sequence Read Archive under accession numbers OR567320–OR568099.

### 2.4. Cytological Examination

Starting from the necrotic lesion present in kidney and liver, smears were prepared, air-dried rapidly, and subsequently stained with Romanowsky stain (May-Grunwald-Giemsa, Merk KGaA, Darmstadt, Germany). At microscopic examination, the sample showed a myriad of budding yeast cell and pseudohyphae exhibiting regular constriction points, immersed in a background of necrosis.

### 2.5. Histopathological Examination

Samples of liver, kidney, lung, spleen, and heart were collected and immersed in buffered 10% formaldehyde solution for fixation. After fixation, the tissues were processed for routine histopathology. Formalin-fixed paraffin-embedded (FFPE) tissues were sectioned in 4 µm thick sections subsequently stained with hematoxylin and eosin (H&E) and Grocott–Gomori’s methenamine silver stain (GMS). Microscopically, multifocal and randomly distributed foci of mineralization surrounded by a small amount of fibrosis and mild lymphoplasmacytic inflammatory infiltrate were detected in the liver. Renal parenchyma was affected by mild, multifocal chronic lymphoplasmacytic tubulointerstitial nephritis; multifocally, tubular lumina were dilated and filled with karyorrhectic and pyknotic neutrophils mixed with macrophages, and multifocal suppurative foci were predominantly observed in the renal medulla.

Grocott–Gomori’s methenamine silver staining revealed myriads of pseudohyphae and narrow-neck budding yeasts, suggestive of *Candida albicans*, mixed with the inflammatory cells described in the renal tubules (Figure 3). Multifocal areas of necrosis of lung parenchyma associated with infiltrates of pyknotic neutrophils and macrophages in the bronchiolar spaces and multifocal perivascular infiltration of lymphocytes were detected. Alveolar spaces were expanded by edema and fibrin. Numerous GMS-positive septate fungal hyphae with parallel walls and dichotomously branching, compatible with *Aspergillus* spp., were observed within a bronchus, mixed with necrotic debris, macrophages, and pyknotic neutrophils (Figure 4). Spleen showed white pulp hyperplasia and the presence of megakaryocytes, while the heart did not show significant findings.

### 2.6. Serum Protein Electrophoresis and Fungal Antigen Detection

The serum was extracted from the clots found in the cardiac chambers to conduct a capillary serum electrophoresis of proteins (Capillarys, Sebia Italia, Bagno a Ripoli, FI, Italy). The total serum proteins were elevated at 9.0 g/dL (reference range: 5.7–7.2 g/dL). The electrophoretic pattern revealed an increase in α-2 fraction at 1.82 (reference range: 0.37–0.7 g/dL) and gamma-globulins at 2.14 g/dL (reference range 0.5–1.5 g/dL), respectively, and an altered A: G ratio of 0.64 (reference range 0.8–1.8).

Additionally, the serum was utilized for the detection of fungal antigens, including *Candida* mannan antigen (Platelia *Candida* Ag Plus #62784—Bio-Rad, Laboratories srl, Segrate- MI, Italy) and galactomannan (Platelia Aspergillus Ag—Bio-Rad Laboratories srl, Segrate- MI, Italy). The first test yielded a positive result, while the second test was negative.

## 3. Discussion

This study describes the occurrence of a mixed infection caused by *Candida albicans* and *Aspergillus fumigatus* in an adult female alpaca that died after prolonged treatment with antibiotic and corticosteroids. In particular, the post-mortem isolation and the microscopic evidence of *Candida albicans* from different tissues and the presence of *Candida* mannan antigen in the serum suggests a systemic infection caused by *C. albicans*. In contrast, the exclusive presence of *A. fumigatus* in the lung and the absence of galactomannan antigen in the serum suggests a localized infection by *A. fumigatus*, which could have occurred subsequently due to the immunosuppression caused by *C. albicans* septicemia and by the intake of drugs favoring colonization by fungi [21].

The clinical history of this case report showed that the animal exhibited respiratory problems during the first veterinary visit. Among fungal infections, the clinical symptoms could be associated to an *Aspergillus* sp. infection rather than a *Candida* sp. infection. Indeed, *A. fumigatus* may cause mycotic pneumonia, gastroenteritis, mastitis, placentitis, and abortions in ruminants, especially cows [34]. *Aspergillus* pneumonia is a fatal disease in ruminants that may progress rapidly [34]. Clinical signs of *Aspergillus* pneumonia in ruminants generally include pyrexia, rapid, shallow, and stertorous respiration, nasal discharge, and a moist cough [34]. On the contrary, *Candida* infections include different clinical presentations varying from relatively benign mucocutaneous diseases to severe, life-threatening invasive infections. The clinical manifestations associated with invasive candidiasis are not pathognomonic and can vary from the presence/absence of fever to sepsis and septic shock [35]. Acute disseminated candidiasis typically presents neutropenic fever that does not respond to antibacterial treatment, accompanied by skin lesions, pneumonia, and shock. Common sites of infection include the skin, lungs, gastrointestinal tract, kidney, liver and spleen [36]. Chronic disseminated candidiasis manifests as a non-neutropenic fever that is unresponsive to antibacterial therapy, and in this type of infection, the spleen and liver are the primary sites of infection [36]. Thus, the initial positive outcome of the alpaca following antibiotic and corticosteroid treatment suggests that fungi were not the cause of the primary clinical signs, although a possible *Aspergillus* spp. colonization cannot be ruled out because of the lack of specific diagnostic procedures.

The subsequent clinical evolution of this case report suggests that the prolonged use of antibiotics and steroids may have favored the infection by both *C. albicans* and *A. fumigatus* in this animal.

*Candida* and *Aspergillus* infections are opportunistic mycoses and the risk factors associated with pathologies may vary according to the fungal spp. In particular, the invasive *Candida* spp. infections can be influenced by various factors, including but not limited to reduced digestive secretions, nutritional deficiencies, dietary factors, compromised immune system, impaired liver function, and a prolonged use of antibiotics and medications [37,38]. Generally, the source of *C. albicans* infection is autogenous and gastrointestinal lesions caused by antibiotic and steroid drugs could have caused its spread to other organs and tissues [21]. On the contrary, the risk factors of invasive pulmonary aspergillosis are usually due to systemic immunosuppression and to an early lactation state [34]. The most likely source of infection is the environment and the transmission route is inhalation, although gastric penetration cannot be excluded in ruminants [34].

In our case, the alpaca was fed good quality hay and pelleted feed and it cannot be ruled out that *Candida* penetration occurred via the digestive tract because of micro-lesions caused by the feed. Unfortunately, no hematological and biochemical tests were performed to aid in understanding the clinical–pathological situation of the patient, apart from the serum electrophoresis performed on the post-mortem sample. The electrophoretic pattern reveals an increase in the alpha-2 and gamma-globulin fraction, typically indicative of an ongoing exacerbation of a chronic inflammatory process.

## 4. Conclusions

Fungal diseases in animals are often insidious, diagnosed late, and sometimes only detected during necropsy. Based on the data reported here, systemic candidiasis and *Aspergillus* pneumonia should be considered as possible differential diagnoses in alpacas presenting respiratory symptoms.

Prolonged treatment with glucocorticoids and antibiotics should be avoided as it could represent a risk factor for the onset of pathologies caused by opportunistic fungi such as *Candida* spp. and *Aspergillus* spp.

## Figures and Tables

**Figure 1 jof-10-00227-f001:**
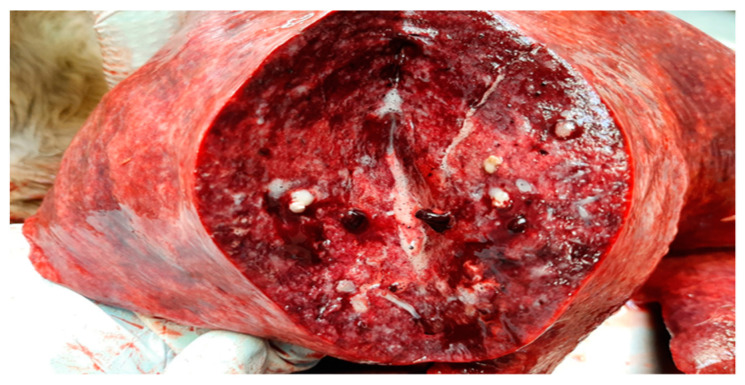
Lung. Moderate, multifocal, necrotizing and suppurative bronchopneumonia with edema.

**Figure 2 jof-10-00227-f002:**
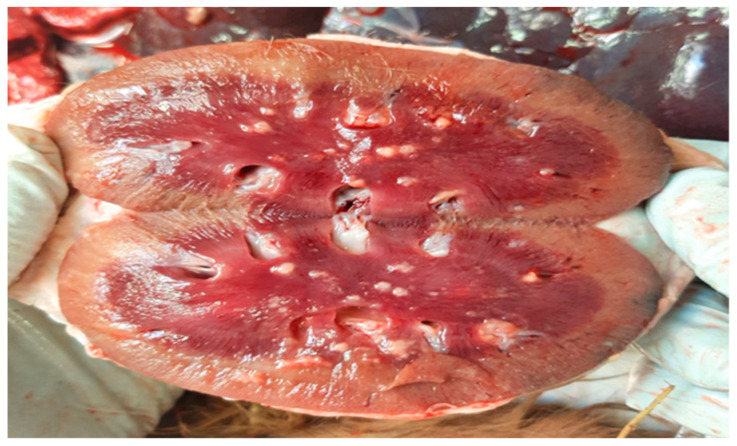
Kidney. Cut surface of kidney showing multifocal to coalescing suppurative foci with predominant involvement of the renal medulla.

**Figure 3 jof-10-00227-f003:**
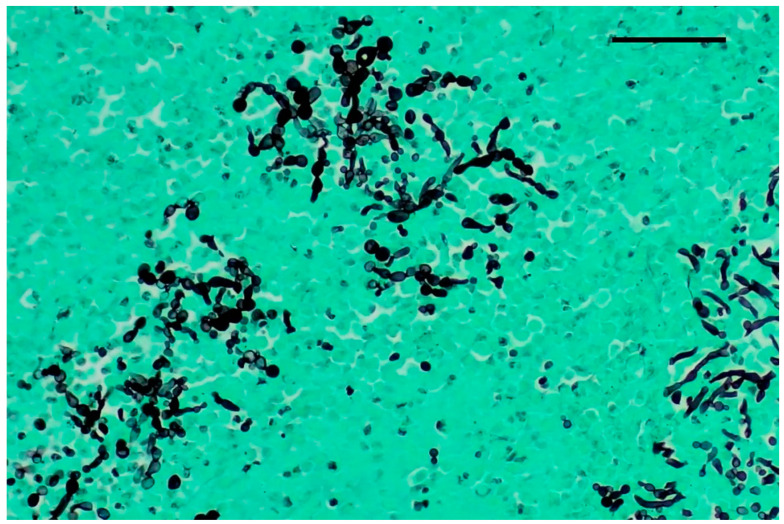
Kidney, Grocott–Gomori’s methenamine silver stain (GMS). Numerous pseudohyphae and narrow-based budding yeasts within renal parenchyma. Original magnification 40×, scale bar: 500 µm.

**Figure 4 jof-10-00227-f004:**
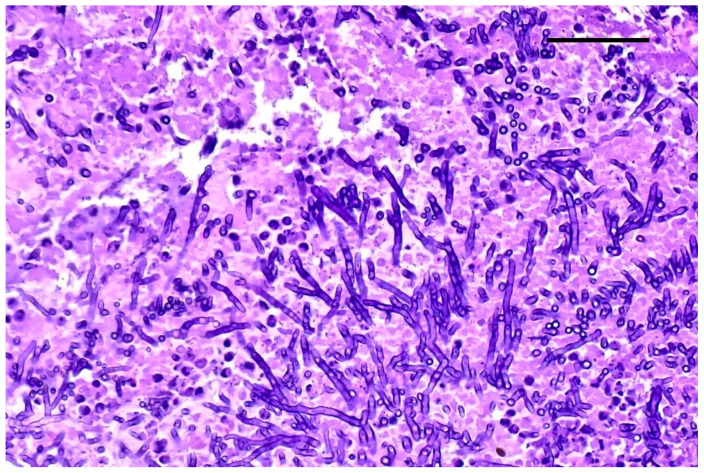
Lung. Fungal hyphae with parallel wall and dichotomous branching; H&E staining, original magnification 40×, scale bar: 500 µm.

## Data Availability

Data are contained within the article.

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
