# Peer review of "Systemic Candida Infection and Pulmonary Aspergillosis in an Alpaca (Vicugna pacos): A Case Report"

_jof, 2024, doi:10.3390/jof10030227_

Round 1

Reviewer 1 Report

Comments and Suggestions for Authors

This is a short case presentation on an unusual co-infection by Candida albicans and Aspergillus in an alpaca. The methodology is sound and the results are well presented. The conclusions are appropriate based on the data provided and the clinical presentation. It will be of interest to veterinarians and clinical mycologists.

Comments on the Quality of English Language

The quality of English is marginal and needs improving. I have made suggestions on the marked-up copy of the paper that is attached. The paper should be read and reviewed by a native English speaker prior to resubmission.

Please also revise the use of paragraphs to improve the structure of the narrative.

Author Response

Your Comment

This is a short case presentation on an unusual co-infection by Candida albicans and Aspergillus in an alpaca. The methodology is sound and the results are well presented. The conclusions are appropriate based on the data provided and the clinical presentation. It will be of interest to veterinarians and clinical mycologists.

Our Comment

The authors thank the reviewer for her/his good consideration and constructive criticisms all the observations in pdf format of the manuscript were accepted and the manuscript has been revised accordingly.

In particular:

  • The sentence “Candida species are widely distributed yeasts that typically benign counter parts of the normal gastrointestinal, urogenital, and cutaneous flora of both humans and warm-blooded animals. In these locations, it primarily exists as budding yeast associated with the mucosal surface [2,3]. More than 200 species of Candida exist, but only a few species have been identified as pathogens [4,5].Candida albicans is commonly identified as the causative agent of candidosis, a condition primarily observed in individuals or animals with compromised immune system or underlying disorders [3,6]. Localized candidosis is familiar in captive wildlife, notably in avian species and mammals [7,8], although less common, disseminated form of the illness may also arise. The dissemination of Candida varies depending on the host’s immune status, and the yeasts can remain localized to the skin and mucous membranes causing a superficial mycosis or can spread through the bloodstream, leading to systemic mycosis [2,9]..” have been changed into: “Candida species are widely distributed yeasts and well recognized as commensal organisms of human/animal skin and /or gastrointestinal and urogenital tracts. The genus comprises more than 200 species, but only few species have been identified as pathogens [2-5].In particular, Candida albicans is commonly identified as the causative agent of candidiasis, a condition observed mainly in human or animals with compromised immune system or underlying disorders [3,6]. Localized candidiasis is common in captive wildlife, notably in avian species and mammals [7,8], whereas disseminated form of the illness are less common described. The dissemination of Candida spp. varies depending on the host’s immune status; the yeasts can remain localized to the skin and mucous membranes causing superficial mycoses in immunocompetent host or can spread through the bloodstream, leading to systemic mycosis in severe immunocompromised host [2,9].”
  • The sentence” Systemic candidiasis is rare in animals, with only few reported cases in dogs [10–13], calves [14], sheep [15,16], cat [17], horses [18], guanaco [19], llama [20], and 2 two cases in alpaca [21]. Has been changed into: “Systemic candidiasis is rarely reported in animals and only few cases were described in dogs [10–13], calves [14], sheep [15,16], cat [17], horses [18], guanaco [19], llama [20] and alpaca [21].”.
  • The sentences” Authors should discuss the results and how they can be interpreted from the perspective of previous studies and of the working hypotheses. The findings and their im-

 plications should be discussed in the broadest context possible. Future research directions may also be highlighted.” has been delated.  The authors apologize for this inaccuracy.

Your comment

The quality of English is marginal and needs improving. I have made suggestions on the marked-up copy of the paper that is attached. The paper should be read and reviewed by a native English speaker prior to resubmission.

Our comments

As for English, the manuscript has been accurately revised by an English speaker from USA who was added in acknowledge section.

Your comments

Please also revise the use of paragraphs to improve the structure of the narrative.

Our Comment

The authors do not understand what the Reviewer means with this comment. The manuscript has been prepared by using paragraphs to make the diagnostic process easier to understand. We follows a recent published manuscript  in JoF (i.e.,  L.-A.; Chuang, Y.-C.; Yeh, T.-K.; Lin, K.-P.; Lin, C.-J.; Liu, P.-Y. "Talaromyces amestolkiae Infection in an AIDS Patient with Cryptococcal Meningitis." J. Fungi 2023, 9, 932. https://doi.org/10.3390/jof9090932).

Reviewer 2 Report

Comments and Suggestions for Authors

This study reports a case of systemic Candida infection associated with pulmonary aspergillosis in an apparently immunocompetent alpaca.  

Please, review the following minor observations:

1)    Please, include the scale in figure 3 and 4.

2)    There is an error in Figure 3, as it has been incorrectly labeled as Figure 2.

3)    Please indicate the reference of the methodology used for the MALDI TOF analyzes. Include identified sequences as supplementary data.

4)    Line 188, A. fumigatus change by A. fumigatus

Author Response

Reply to Reviewer #2

Your Comments

This study reports a case of systemic Candida infection associated with pulmonary aspergillosis in an apparently immunocompetent alpaca.  

Please, review the following minor observations:

1)  Please, include the scale in figure 3 and 4.

2)  There is an error in Figure 3, as it has been incorrectly labeled as Figure 2.

3) Please indicate the reference of the methodology used for the MALDI TOF analyses. Include identified sequences as supplementary data.

4)   Line 188, A. fumigatus change by A. fumigatus

Our Comments

The authors thank the reviewer his/her suggestions. The manuscript has been revised accordingly.

As for the “the methodology used for the MALDI TOF analyses” the following sentence has been added in the revised version of the manuscript:

“The yeast and filamentous colonies were identified with high confidence (range >2) as Candida albicans and Aspergillus fumigatus, respectively, using a mass spectrometer MALDI-TOF Biotyper® (Bruker Daltonics GmbH & Co.KG., Bremen, Germany), following the manufacturer's instructions for fungi.”